# Serum Chemerin Concentration Is Associated with Proinflammatory Status in Chronic Coronary Syndrome

**DOI:** 10.3390/biom11081149

**Published:** 2021-08-04

**Authors:** Anna Szpakowicz, Malgorzata Szpakowicz, Magda Lapinska, Marlena Paniczko, Slawomir Lawicki, Andrzej Raczkowski, Marcin Kondraciuk, Emilia Sawicka, Malgorzata Chlabicz, Marcin Kozuch, Maciej Poludniewski, Slawomir Dobrzycki, Irina Kowalska, Karol Kaminski

**Affiliations:** 1Department of Cardiology, Medical University of Bialystok, ul.Jana Kilinskiego 1, 15-089 Białystok, Poland; akodzi@poczta.onet.pl (A.S.); emiliasawickak@gmail.com (E.S.); 2Department of Population Medicine and Lifestyle Diseases Prevention, Medical University of Bialystok, ul.Jana Kilinskiego 1, 15-089 Białystok, Poland; malgorzata.szpakowicz@umb.edu.pl (M.S.); magda.lapinska@umb.edu.pl (M.L.); marlena.paniczko@umb.edu.pl (M.P.); slawomir.lawicki@umb.edu.pl (S.L.); andrzej.raczkowski@umb.edu.pl (A.R.); marcin.kondraciuk@umb.edu.pl (M.K.); malgorzata.chlabicz@umb.edu.pl (M.C.); 3Department of Invasive Cardiology, Medical University of Bialystok, ul.Jana Kilinskiego 1, 15-089 Białystok, Poland; marcin.kozuch@umb.edu.pl (M.K.); mpoludniewski@yahoo.com (M.P.); slawomir.dobrzycki@umb.edu.pl (S.D.); 4Department of Internal Medicine and Metabolic Diseases, Medical University of Bialystok, ul.Jana Kilinskiego 1, 15-089 Białystok, Poland; irina.kowalska@umb.edu.pl

**Keywords:** chemerin, coronary artery disease, body composition, white blood cells count, neutrophil to lymphocyte ratio, insulin resistance

## Abstract

Background: Chemerin is an adipokine and a chemoattractant for leukocytes. Increased chemerin levels were observed in patients with coronary artery disease (CAD). We investigated associations between chemerin and biochemical measurements or body composition in CAD patients. Methods: In the study, we included patients with stable CAD who had undergone percutaneous coronary intervention (PCI) in the past. All patients had routine blood tests, and their insulin and chemerin serum levels were routinely measured. Body composition was assessed with the DEXA method. Results: The study group comprised 163 patients (mean age 59.8 ± years, 26% of females, n = 43). There was no significant difference in serum chemerin concentrations between patients with diabetes and the remaining ones: 306.8 ± 121 vs. 274.15 ± 109 pg/mL, *p* = 0.1. Chemerin correlated positively with the white blood cell (WBC) count, the neutrophil to lymphocyte ratio, hsCRP, all fractions of cholesterol, triglycerides, platelet count, fasting insulin, and c-peptide. Chemerin levels were also correlated with total fat mass but only in a subgroup with normal glucose metabolism. Conclusion: In patients with CAD, serum chemerin levels are correlated with inflammation markers, insulin resistance, and an unfavorable lipid profile. Correlation with fat mass is dependent on glucose metabolism status. Depending on the presence of diabetes/prediabetes, the mechanisms regulating chemerin secretion may be different.

## 1. Background

### 1.1. Mechanisms of Chemerin Action

Chemerin is a protein that plays dual role as a chemoattractant for leukocyte populations and as an adipokine regulating lipid and carbohydrate metabolism and thus exerts proinflammatory action and promotes insulin resistance [1,2]. It binds to chemokine-like receptor 1 (CMKLR1), G protein-coupled receptor 1 (GPR1), and C-C motif receptor-like 2 (CCRL2) [3]. Chemerin mediates its functions mainly via the CMKLR1 receptor, which is present mainly on dendritic cells, monocytes/macrophages, and adipocytes [3,4,5,6] (Figure 1). The GPR1 receptor has no prominent effects that have been described, and the CCRL2 receptor does not mediate any intracellular mediating pathways and is considered a decoy receptor [3] (Figure 1).

### 1.2. Chemerin in Clinics

Serum concentrations of chemerin is increased in metabolic syndromes and corresponds well with components of metabolic syndrome phenotypes: obesity, adverse lipid profile, increased blood pressure, diabetes and prediabetes, and hepatic steatosis [1,7,8,9,10]. It contributes to several diabetes complications: diabetic nephropathy, retinopathy, or coronary artery disease (CAD) [2,11,12,13,14,15]. It is elevated in patients with dilated cardiomyopathy [16]. It was strongly associated with a risk of heart failure in a prospective long-term observation in an EPIC-Potsdam cohort [17], and it corresponds to an adverse outcome in heart failure patients [18]. Empagliflozin treatment reduces chemerin mRNA levels in the hearts of Zucker diabetic fatty rats [19]. Chemerin also promotes pulmonary artery vasoconstriction in rats [20]. Other observations concerning this adipokine include inverse correlations with the estimated glomerular filtration rate (eGFR) in the general population and with bone quality in obese patients [21,22].

On the other hand, chemerin may also have some other effects: it increases eNOS activity and is pro-angiogenic [23,24]. Furthermore, it is inversely associated with the coronary artery calcium score in patients with chronic kidney disease [25]. Therefore its role in CAD and atherosclerosis is not fully understood. In general, chemerin concentrations are elevated in patients with CAD or stroke [2,14,15,26]. Additionally, a large prospective study revealed a strong positive association between chemerin levels and the prevalence of those diseases [27]. Serum chemerin concentrations correspond with a number of atherosclerotic lesions in coronary arteries, their stability, and patient prognosis [2,15,28]. These data are not supported by results concerning subclinical parameters of atherosclerosis. In the SHIP study, which was conducted in the general population, no association was observed between chemerin and pulse wave velocity (PWV) or carotid ultrasound [29]. In this study only a modest inverse correlation was reported for the ankle-brachial index (ABI) [29].

### 1.3. Aims of the Study

In this pilot study, we investigated associations between serum chemerin levels and biochemical measurements (including insulin resistance indices), body composition, and the condition of large arteries (PWV, carotid ultrasound, ABI). This research was performed in patients with angiographically confirmed CAD, while most of previous studies concerned the general population or patients with diabetes.

## 2. Material and Methods

### 2.1. Study Population

In this observational study, we included 163 patients with chronic coronary syndrome. They all underwent percutaneous coronary intervention (PCI) in the past (median 16 months) and had no anginal pain after the procedure. Additionally, we formed a control group of 34 individuals without CAD and excluded diabetic and prediabetic patients who were gender and age matched to the subgroup of patients with CAD and normal glucose tolerance. Patients from the control group were recruited from the Bialystok PLUS study [30].

CAD was excluded based on patient history, including the lack of anginal pain. Diabetes and prediabetes were ruled out on the basis of history and/or oral glucose tolerance test (OGTT). According to study design, all of the procedures and blood tests would be performed during one meeting, therefore we only had one fasting glucose measurement [30]. OGTT was only performed in individuals without a history of diabetes, irrespective of the fasting glucose level that was unavailable at the time the test began. Diabetes was diagnosed in patients with a glucose level at 2 h in OGTT ≥ 200 mg/dL. Prediabetes, impaired fasting glucose (IFG), and impaired glucose tolerance (IGT) were diagnosed as follows: IFG was diagnosed in patients with both fasting glucose levels 100–125 mg/dL and a glucose level at 2 h in OGTT < 140 mg/dL, and IGT was diagnosed in patients with a glucose level at 2 h in OGTT between 140–199 mg/dL. Glucose metabolism was considered normal if the fasting glucose level was <100 mg/dL and the glucose level at 2 h in OGTT was <140 mg/dL. There were no patients with unclear diabetic status in the study group.

### 2.2. Performed Procedures

Participants were asked about previous myocardial infarction, coronary revascularization, and chest pain. Patients with typical anginal chest pain (according to a physician’s assessment) or with left ventricular ejection fraction ≤ 50% (Simpson method), were not recruited to the control group. Serum chemerin levels were measured with the ELISA method (Biovendor), according to manufacturer’s instructions. All patients had routine blood tests and had their insulin levels measured. Patients without a history of diabetes also had OGTT. Routine biochemical tests were performed in the Cobas c111 machine (Roche). The morphological parameters of the blood were assessed with the Mythic 18 machine (Cormay). Insulin and c-peptide were measured with the Cobas e411 machine (Roche). The homeostasis model assessment of insulin resistance (HOMA-IR) was only calculated in patients without diabetes, according to the equation: fasting insulin µU/mL) × fasting glucose (mmol/L)/22.5. The status of large the arteries was evaluated with carotid ultrasound, pulse wave velocity (PWV, Vascular Explorer device, GmbH, Dusseldorf, Germany), and the ankle brachial index (ABI). Ultrasound was performed with the Vivid™9 device (GE Healthcare, Chicago, IL, USA). We analyzed the presence of atherosclerotic plaques and intima-media thickness (IMT). We compared chemerin levels in patients with the following measurements: IMT > 0.9 mm and ≤0.9 mm; PWV > 10 m/s and ≤10 m/s; and ABI < 0.9 and ≥0.9. Body composition was assessed using the dual-energy X-ray absorptiometry method (Lunar iDXA, GE Healthcare, Chicago, IL, USA). The anatomical severity of CAD was evaluated with a SYNTAX score in angiographies performed during previous PCI 6–18 months prior to enrolment to the study [31].

### 2.3. Statistical Analysis

Variable distribution was assessed with the Shapiro–Wilk test. Depending on distribution, groups were compared with the nonparametric Mann–Whitney U test or the parametric Student’s *t*-test. Categorical variables were compared with the chi-square test. The correlation between chemerin levels and other parameters was calculated with the Spearman test and was presented as Spearman rank order correlations (R) with *p* values. Associations between chemerin and laboratory findings were verified for dependence on BMI in multiple regression. Multivariable analysis concerning parameters independently associated with chemerin levels was performed with multiple regression. The primary model included WBC count, platelet count, android fat mass, gynoid fat mass, lean body mass, diagnosis of diabetes or prediabetes, and c-peptide concentration. Variables were eliminated from the model in a backward stepwise manner. All calculations were performed with STATISTICA software.

### 2.4. Ethical Statement

The protocol of the study was approved by the Medical University of Bialystok Bioethics Committee (ID of the approval R-I-002/323/2016). The study was performed in accordance with the ethical standards put forward in the 1964 Declaration of Helsinki and its later amendments. Informed written consent was obtained from all participants prior to their inclusion in the study.

## 3. Results

### 3.1. Clinical Characteristics of the Studied Groups

In the study, we included 163 patients with CAD and a mean age 59.8 ± 6 years, 26% of whom were women (n = 43). General characteristics of the study group are shown in Table 1.

A total of one hundred and twenty-nine patients had type 2 diabetes or prediabetes (79.1%, Group 1), and the remaining 34 had normal glucose tolerance (Group 2). Additionally, we investigated 35 participants without CAD and with normal glucose control (Group 3) who were age and sex matched to Group 2.

Table 2 presents the clinical characteristics of the studied groups. Patients from Group 1 (CAD with diabetes or prediabetes) compared to Group 2 (CAD and normal glucose tolerance) had significantly higher BMI, fat mass, insulin resistance parameters, neutrophil to lymphocyte ratio (NLR), and hsCRP levels (Table 2). Patients from Group 3 (no CAD and normal glucose tolerance) had significantly higher total cholesterol and LDL cholesterol levels compared to Group 2. In Group 3, only 2 out of 35 patients were on statins due to hypercholesterolemia.

### 3.2. Serum Chemerin Concentrations, Body Composition and Biochemical Parameters

The mean serum chemerin level in the whole study group was 284.8 ± 113 ng/mL. The difference between the patients from the investigated groups was not significant (Table 2). In patients with CAD, chemerin correlated positively with android and total fat mass, c-peptide, fasting insulin, acute phase markers (all fractions of leukocytes, NLR, and hsCRP), platelet count, and adverse lipid profile (Table 3). Only patients with CAD and normal glucose tolerance (Group 2) presented a correlation between chemerin and hsCRP or fat mass (total, gynoid and android fat mass as well as percentage of fat tissue) (Table 3). On the other hand, a correlation between chemerin and leukocytes or NLR was only observed in patients with diabetes/prediabetes (Group 1). There were no significant correlations in Group 3.

Through multiple regression, we verified whether associations between chemerin and the laboratory findings were dependent on BMI (Table 4). In the whole group of patients with CAD, after adding BMI to the multiple regression model, chemerin was no longer associated with HDL cholesterol, LDL cholesterol, hsCRP, or lymphocyte count. In Group 1, a significant correlation between WBC count, monocyte count, and granulocyte count remained. In Group 2, a significant correlation remained for platelet count and triglycerides.

In the multivariable analysis of all of the patients with CAD, three parameters were independently associated with chemerin levels: WBC count (β = 0.23, *p* = 0.002), android fat mass (β = 0.18, *p* = 0.01), and platelet count (β = 0.17, *p* = 0.03), (R2 = 0.12, *p* < 0.001). The primary model additionally included android fat mass, gynoid fat mass, lean body mass, the diagnosis of diabetes or prediabetes, and c-peptide concentration, which were eliminated from the model in a backward stepwise manner.

### 3.3. Serum Chemerin Concentrations and Large Arteries

Within the group CAD, 93 patients had carotid artery atherosclerosis (57%). Chemerin levels did not differ significantly between patients with and without carotid atherosclerosis (300.7 ± 124 vs. 263.6 ± 95 ng/mL, *p* = 0.07, Mann–Whitney U test). Furthermore, serum chemerin concentration was associated neither with IMT (292.3 ± 102 ng/mL in patients with IMT > 0.9 mm (n = 46) vs. 274.9 ± 112 ng/mL in patients with IMT ≤ 0.9 mm, *p* = 0.34) nor with PWV (295.4 ± 111 ng/mL in patients with PWV > 10 m/s (n = 27) vs. 288.3 ± 106 ng/mL in patients with PWV ≤ 10 m/s, *p* = 0.75). A total of eight patients had an ankle brachial index (ABI) < 0.9 and a mean serum chemerin level of 295.9 ± 137 ng/mL compared to 276.6 ± 109 ng/mL in patients with ABI ≥ 0.9, *p* = 0.4. The mean SYNTAX score in the study group was 11.1 ± 8.2. It did not correlate with chemerin levels.

## 4. Discussion

In our study performed in patients with chronic coronary syndromes, serum chemerin levels were associated with proinflammatory status, insulin resistance, and an unfavorable lipid profile.

We observed the association between serum chemerin and white blood cell count and NLR and CRP in patients with CAD. It should be underlined that a correlation between chemerin and leukocytes or NLR was only observed in patients with diabetes/prediabetes. NLR is a marker of subclinical inflammation and a predictor of all-cause mortality and cardiovascular events in patients with CAD [32]. In our study, a correlation between chemerin and NLR, however, was dependent on BMI. After adjustment for BMI, chemerin remained related to leukocytes, monocytes, and granulocytes. It is well recognized that the accumulation of macrophages and other immune cells in atherosclerotic plaque promotes disease progression. There is only one study presenting similar results. Er et al. noticed that in patients with confirmed angiographically CAD, the serum chemerin level was related to leukocyte count and CRP [28]. However, they did not mention the diabetic status of the patients. This observation could argue for the significance of the proinflammatory action of chemerin in plaque formation and the progression of atherosclerosis. To support the chemerin role as a chemoattractant protein in the progression of atherosclerosis, Kostopoulos et al. showed that the local expression of chemerin and its receptor CMKLR1 in pericoronary and periaortic adipose tissue is related to the severity of atherosclerosis [33]. Thus, circulating chemerin as well as locally secreted chemerin could contribute to disease progression.

We observed a positive correlation between serum chemerin and the serum concentration of insulin and c-peptide in patients with CAD, which persisted after adjustment for BMI. Similar results have been obtained by Yan et al. in Chinese CAD patients [2]. It has been reported that hyperinsulinemia increases serum chemerin level [34]. However, the data about chemerin and insulin sensitivity are inconsistent. In vitro studies on 3T3-L1 adipocytes provide conflicting results regarding glucose uptake by cultured adipocytes [35,36]. In skeletal muscle cells, chemerin inhibited glucose uptake [37]. Clinical studies have shown the correlation of serum chemerin with indirect indices of insulin resistance [38,39]. Our data obtained in patients with CAD are in agreement with quoted studies, as fasting insulin could be also considered as an indirect measure of insulin resistance.

In our study, patients from the control group had a significantly higher total and LDL cholesterol levels compared to patients with CAD. This can be explained by routine statin use in patients with CAD (almost 90%). Despite that, the mean LDL level in this group was far above the target value (93.9 mg/dL, target value at the time of the study <70 mg/dL). In the large SHIP study, a significant association between chemerin and all fractions of the routine lipid profile (triglycerides, total, LDL and HDL cholesterol) was reported in the general population as well as in healthy subjects [7]. We confirmed this finding in our group of patients with CAD. Some previous studies in CAD patients already reported the association between chemerin and lipid profile but did not include HDL cholesterol [2,8]. It should be noted that this is the first study to show this phenomenon. The observed correlation, however, was dependent on BMI influence and was lost after adjustment in a multivariate model. In our study, no correlation between lipids and chemerin was found in the control group, probably due to the small number of patients included.

In our group of patients with CAD, chemerin correlated with platelet count, which is consistent with the study performed by Er et al. [28]. Such a phenomenon was not reported for the general population.

Our results do not confirm the data obtained from previous studies concerning an association between chemerin levels and CAD [2,14,15,27]. The possible explanation for our observations of lacking differences between CAD and control patients could be the aspirin use in almost all our patients with CAD. Herova et al. showed that a small dose aspirin could be responsible for lower chemerin concentrations in patients with CAD. Patients not taking aspirin had significantly higher chemerin concentrations [9].

Li Q et al. reported that chemerin is a biomarker of an acute coronary syndrome [15]. Its association with chronic coronary syndromes was not confirmed in this study despite only 40% of patients having chronic coronary syndromes and 15% of controls being treated with aspirin [15]. The probable explanation for those results would be thr small number of cases and the controls included in the analysis (n = 60 each group) [15].

Yan Q et al. observed that in patients undergoing scheduled coronary angiography, chemerin levels were higher in patients with confirmed CAD compared to non-CAD patients [2]. The number of patients treated with acetylsalicylic acid, however, was not reported in this study [2]. We can assume that it would be significantly lower than it was in our population because at the time of enrolling in the study, most of the patients had not yet been diagnosed as having CAD.

In a large prospective EPIC study, serum chemerin levels were measured in 254 participants who developed myocardial infarction (MI) during follow-up and in 2342 participants with excluded cardiovascular disease [27]. In a multivariate model adjusted for multiple cardiovascular risk factors (including diabetes), a strong positive association between chemerin and MI was reported: hazard ratio quartile 4 vs. quartile 1 (HR): 3.13 (95% CI 1.91–5.11). An even stronger correlation was reported for chemerin and diabetes (HR = 3.57; 95% CI: 1.75–7.28), however, it was markedly attenuated by the adjustment for body mass index and waist circumference (HR 1.70; 95% CI: 0.83–3.50) [27].

So far, there are no dedicated studies that have assessed the relation between chemerin and CAD depending on diabetic status. Additionally, studies focused on the mechanism of this interaction do not perform subanalyses based on differences in glucose tolerance status. Only Lin X et al. exclusively investigated patients with type 2 diabetes undergoing scheduled angiography and reported that patients with confirmed CAD had higher chemerin levels than patients without CAD [14]. Patients with normal glucose metabolism or prediabetes were not included in this study [14]. Yan Q et al. performed a subanalysis in patients with CAD depending on their metabolic syndrome status, and no differences in serum chemerin levels were found [2].

In our patients with CAD, chemerin levels correlated with fat mass, android fat mass, and gynoid fat mass but only in the subgroup with normal glucose metabolism. No correlations were observed in the patients with CAD and diabetes/prediabetes. It is a surprising finding that suggests that depending on presence of diabetes/prediabetes, the mechanisms regulating the secretion of chemerin are different. This hypothesis is supported by a large study performed in 1215 patients with vascular disease [40]. They reported that adiposity (which is strongly related to glucose metabolism disorders) was associated with lower serum chemerin levels.

We did not observe an impact of chemerin on the condition of the large arteries in patients with CAD. In the SHIP trial, analysis of carotid ultrasound and ABI was performed in four thousand participants representing the general population [29]. A significant association with chemerin was observed in the case of ABI, carotid plaque, carotid stenosis, and IMT. After adjustment for confounders, only ABI remained inversely and significantly correlated [29]. Another trial confirmed a correlation between chemerin and carotid atherosclerosis or carotid plaque stability [26]. Several studies reported an association between chemerin and CAD severity assessed with a Gensini score [2,15,41]. Chemerin concentration was also a risk factor for multiple vessel disease [6,28]. According to the guidelines of the European Society of Cardiology [42], we assessed the complexity of CAD with the SYNTAX score and found no correlation.

Many of chemerin correlations observed in this study are considered weak (R < 0.4). We did not expect them to be strong because atherosclerosis, insulin resistance, or inflammation are very complex processes that are influenced by multiple factors. Weak correlations were especially observed in the case of the summarized analysis for Groups 1 and 2. When the groups were analyzed separately, the R correlation coefficients were markedly higher.

Some of surprising results of our study could potentially be explained by a novel interesting parameter: the chemerin/adiponectin ratio. Unfortunately, adiponectin concentrations were not measured in our study. In previous reports, high values of this ratio were associated with metabolic syndrome or with dyslipidemia [43].

## 5. Limitations of the Study

The main limitations of our study are the small number of patients included and the observational characteristics using only a single measurement of chemerin.

We are not able to decisively comment on the involvement of chemerin in the development of atherosclerosis since we included patients with established coronary artery disease, and the severity of CAD was assessed approximately a year prior to the other assessments. Invasive coronary angiography was not repeated nor was coronary computed tomography angiography performed at the time of study assessments because of ethical concerns. There were no clinical indications for the need of it, and the procedures are associated with risk.

Neither type of coronary angiography was performed in the control group; therefore, CAD could not be excluded with certainty.

Our study group comprised patients with obstructive CAD who had undergone PCI in the past, so our results and conclusions cannot simply be extended to patients with CAD and insignificant atherosclerotic lesions that are not qualified for revascularization. Moreover, it is also not representative for subclinical atherosclerosis.

## 6. Conclusions

Serum chemerin levels were associated with inflammation markers (including WBC count and NLR), insulin resistance, and an unfavorable lipid profile. Additionally, chemerin levels were positively correlated with fat mass, but only in the subgroup with normal glucose metabolism. These observations indicate that depending on the presence of diabetes/prediabetes, the mechanisms regulating the secretion of chemerin may be different. Chemerin by affecting inflammation and the regulation the glucose and lipid metabolism might be involved in the pathogenesis of atherosclerosis in the early stages of the disease.

## Figures and Tables

**Figure 1 biomolecules-11-01149-f001:**
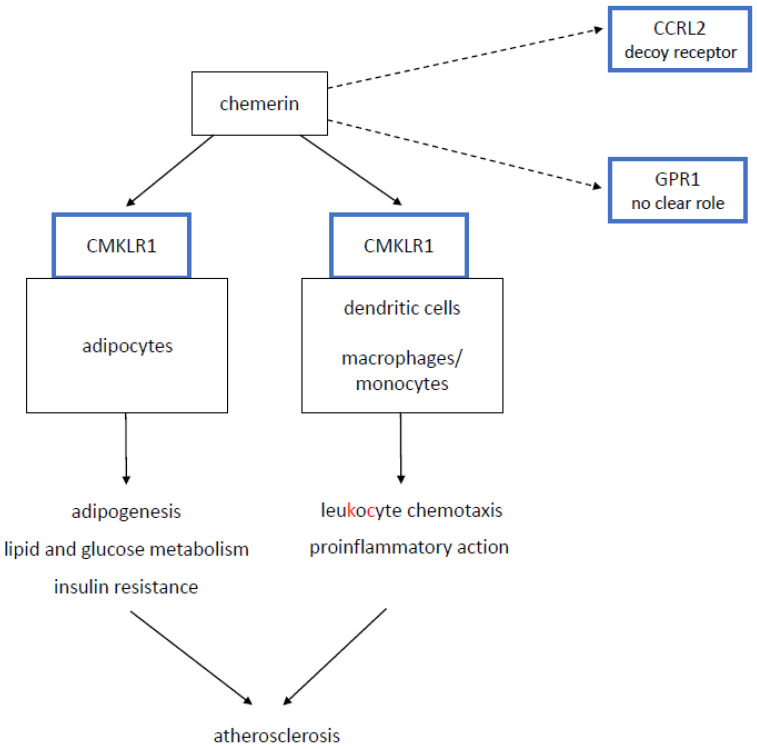
Basic patophysiological pathways of chemerin. CMKLR1—chemokine-like receptor 1; GPR1—G protein-coupled receptor 1; CCRL2—C-C motif receptor-like 2. Solid line—major pathways. Dotted line—minor pathways.

**Table 1 biomolecules-11-01149-t001:** Baseline characteristics of the study group.

Variable	n = 163
Age (years)	59.8 ± 6.0
Female gender (%)	26 (n = 43)
BMI (kg/m^2^)	30.35 ± 5.6
Systolic blood pressure (mmHg)	132 ± 19.4
Diastolic blood pressure (mmHg)	84.2 ± 10.8
Heart rate (beats/min)	66.4 ± 10.7
Hypertension (%)	77.9 (n = 127)
Diabetes (%)	32.5 (n = 53)
Current smoking (%)	25.1 (n = 41)
LDL cholesterol (mg/dL)	93.9 ± 38.6
Carotid artery disease (%)	57 (n = 93/163)
Acetylsalicylic acid (%)	85.3 (n = 139)
Statin treatment (%)	89.6 (n = 146)
Fibrate treatment (%)	4.9 (n = 8)
ACE-I/sartan treatment (%)	87.7 (n = 143)
Beta-blocker (%)	88.9 (n = 145)
Insulin (%)	5.5 (n = 9)
Oral antidiabetic agents (%)	28.2 (n = 46)
Chemerin (ng/mL)	284.8 ± 113

**Table 2 biomolecules-11-01149-t002:** Clinical characteristics of the studied groups.

Characteristic	Group1 (CAD & Diabetes or Prediabetes) n = 129	Group 2 (CAD & Normal Glucose) n = 34	Group 3 (No CAD & Normal Glucose) n = 35	P1 (Group 1 vs. 2)	P2 (Group 2 vs. 3)
Age (years)	60.3 ± 5.5	57.9 ± 7.2	55.9 ± 9.4	0.09	0.38
Female gender (%)	24 (n = 31)	35.3 (n = 12)	34.3 (n = 28)	0.18	0.93
BMI (kg/m^2^)	31.3 ± 5.5	26.8 ± 4.5	26 ± 3.5	0.0001 *	0.38
Fat mass (kg)	32 ± 9.9	23.7 ± 7.5	23.8 ± 7.4	<0.0001 *	0.95
% fat mass (%)	33 ± 6	29.6 ± 6	31 ± 6.7	0.001	0.53
Gynoid fat mass (kg)	4.3 ± 1.5	3.3 ± 1.1	3.5 ± 1.1	0.0007	0.49
Android fat mass (kg)	3.6 ± 1.3	2.4 ± 1.1	2.2 ± 0.9	<0.0001	0.42
Fasting glucose (mg/dL)	119.3 ± 41.3	90.2 ± 6.8	92.1 ± 5.2	<0.0001	0.33
2-h OGTT glucose (mg/dL)	148.7 ± 41.7 (n = 85) ^	97.5 ± 21.7	105.3 ± 19	<0.0001	0.14
Fasting insulin (mU/mL)	18.8 ± 15.9	10.5 ± 5.6	8.9 ± 3.6	<0.0001	0.48
c-peptide (ng/mL)	7.3 ± 40.3	3.2 ± 1.7	2.1 ± 0.5	0.01	0.001
HOMA-IR	4.4 ± 3.3 (n = 76) ^#^	2.36 ± 1.3	2.0 ± 0.8	<0.0001	0.78
White blood count (thous/µL)	6.65 ± 1.7	6.7 ± 2.3	5.6 ± 1.4	0.65	0.049
Neutrophil to lymphocyte ratio	2.4 ± 0.9	2.1 ± 0.6	2.0 ± 0.7	0.04	0.7
Hemoglobin (g/dL)	14.1 ± 1.1	14.2 ± 1.3	14.5 ± 1.2	0.61	0.49
Platelets (thous/µL)	221 ± 55	235 ± 78	232 ± 62	0.96	0.6
hsCRP (mg/L)	2.2 ± 2.7	1.3 ± 2.3	1.4 ± 1.7	0.02	0.99
Total cholesterol (mg/dL)	165.3 ± 49	153. 7 ± 37	210.8 ± 38	0.35	<0.0001
LDL cholesterol (mg/dL)	95.7 ± 40.2	87.3 ± 31.5	142.5 ± 31	0.49	<0.0001
HDL cholesterol (mg/dL)	50.1 ± 15	53.2 ± 18	58.5 ± 16.3	0.34	0.14
Triglycerides (mg/dL)	136.2 ± 87	104 ± 47	115 ± 50	0.064	0.33
Chemerin (ng/mL)	289.5 ± 116	266.8 ± 102	260.3 ± 85	0.37	0.96

* Normal distribution, *t*-test used; ^ OGTT only performed in patients without previous history of diabetes; ^#^ only calculated in patients with prediabetes.

**Table 3 biomolecules-11-01149-t003:** Spearman rank order correlations for chemerin.

Characteristic	Group1+2 n = 163	Group 1 (CAD & Diabetes or Prediabetes) n = 129	Group 2 (CAD & Normal Glucose) n = 34	Group 3 (No CAD & Normal Glucose) n = 35
Fat mass (kg)	R = 0.16 *p* = 0.047	NS	R = 0.41 *p* = 0.016	NS
% fat mass (%)	NS	NS	R = 0.37 *p* = 0.035	NS
Gynoid fat mass (kg)	NS	NS	R = 0.42 *p* = 0.013	NS
Android fat mass (kg)	R = 0.17 *p* = 0.036	NS	R = 0.42 *p* = 0.015	NS
Fasting glucose (mg/dL)	NS	NS	NS	NS
2-h GTT glucose (mg/dL) ^#^	NS	NS	NS	NS
Fasting insulin (mU/mL)	R = 0.22 *p* = 0.004	R = 0.17 *p* = 0.048	R = 0.49 *p* = 0.008	NS
c-peptide (ng/mL)	R = 0.27 *p* = 0.0005	R = 0.26 *p* = 0.003	R = 0.36 *p* = 0.035	NS
HOMA-IR *	NS	NS	R = 0.45 *p* = 0.007	NS
White blood count (thous/µL)	R = 0.34 *p* < 0.0001	R = 0.36 *p* < 0.001	NS	NS
Neutrophil to lymphocyte ratio	R = 0.16 *p* = 0.03	R = 0.18 *p* = 0.04	NS	NS
Lymphocyte count (thous/µL)	R = 0.2 *p* = 0.01	R = 0.19 *p* = 0.03	NS	NS
Monocyte count (thous/µL)	R = 0.23 *p* = 0.003	R = 0.25 *p* = 0.003	NS	NS
Granulocyte count (thous/µL)	R = 0.32 *p* < 0.0001	R = 0.33 *p* = 0.001	NS	NS
Hemoglobin (g/dL)	NS	NS	NS	NS
Platelets (thous/µL)	R = 0.23 *p* = 0.002	R = 0.2 *p* = 0.02	R = 0.4 *p* = 0.018	NS
hsCRP (mg/L)	R = 0.16 *p* = 0.03	NS	R = 0.37 *p* = 0.032	NS
Total cholesterol (mg/dL)	R = 0.17 *p* = 0.03	R = 0.15 *p* = 0.08	NS	NS
LDL cholesterol (mg/dL)	R = 0.19 *p* = 0.01	NS	NS	NS
HDL cholesterol (mg/dL)	R = −0.21 *p* = 0.006	R = −0.18 *p* = 0.044	NS	NS
Triglycerides (mg/dL)	R = 0.3 *p* < 0.0001	R = 0.27 *p* = 0.002	R = 0.38 *p* = 0.024	NS
Syntax score	NS	NS	NS	NA

^#^ Only performed in patients without history of diabetes; * only calculated in patients without diabetes; NS—not significant, NA—not available.

**Table 4 biomolecules-11-01149-t004:** Multiple regression for associations between chemerin and other laboratory measurements adjusted for BMI.

Characteristic	Group1+2 n = 163	Group 1 (CAD & Diabetes or Prediabetes) n = 129	Group 2 (CAD & Normal Glucose) n = 34
Fasting insulin (mU/mL)	β = 0.24 *p* = 0.0007	NS	NS
c-peptide (ng/mL)	β = 0.18 *p* = 0.03	NS	NS
HOMA-IR *	NA	NA	NS
White blood count (thous/µL)	β = 0.26 *p* = 0.003	β = 0.2 *p* = 0.015	NA
Neutrophil to lymphocyte ratio	NS	NS	NA
Lymphocyte count (thous/µL)	NS	NS	NA
Monocyte count (thous/µL)	β = 0.2 *p* = 0.02	β = 0.2 *p* = 0.036	NA
Granulocyte count (thous/µL)	β = 0.26 *p* = 0.002	β = 0.22 *p* = 0.02	NA
Platelets (thous/µL)	β = 0.23 *p* = 0.005	NS	β = 0.6 *p* = 0.004
hsCRP (mg/L)	NS	NA	NS
Total cholesterol (mg/dL)	β = 0.17 *p* = 0.04	NS	NA
LDL cholesterol (mg/dL)	NS	NA	NA
HDL cholesterol (mg/dL)	NS	NS	NA
Triglycerides (mg/dL)	β = 0.2 *p* = 0.014	NS	β = 0.43 *p* = 0.04

* Only calculated in patients without diabetes; NS—not significant, NA—not available (not calculated). β—standardized regression coefficient.

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
