# Peer review of "Serum Chemerin Concentration Is Associated with Proinflammatory Status in Chronic Coronary Syndrome"

_biomolecules, 2021, doi:10.3390/biom11081149_

Round 1
Reviewer 1 Report
See the attached file.

Author Response
Comments:
- From a clinical standpoint, this study is largely negative, and likely not the expected outcome when the investigators began this study, i.e., no difference in chemerin between CAD with diabetes, CAD and normal glucose, and no CAD with normal glucose.
Thank you for this crucial remark. We indeed did not expect such results. However, the study was not designed to investigate the association between chemerin and CAD/diabetes prevalence. We aimed to search for potential associations between chemerin and other various parameters (biochemistry, body composition) that could partially explain previously described link between chemerin and CAD/diabetes. Another possible explanation for our observations of lacking differences between CAD and control patients could be the aspirin use in almost all our patients with CAD. Herova et al. showed that small dose aspirin could be responsible for lower chemerin concentrations in patients with CAD. Patients not taking aspirin had significantly higher chemerin concentrations [7]. (lines 254, 256)
- In the discussion, the authors have detailed previous reports regarding the physiologic and pathophysiologic role of chemerin in metabolic syndrome. This field is in its infancy with many different observations, with some reports contradicting others.
We agree with your comment that reports concerning chemerin action are incomplete and, in some part, inconsistent. Taking into account great clinical impact of this molecule, this was the reason for us to start research in this field.
- The investigators have been able to link serum chemerin concentrations with other inflammatory marker which may offer some insights. Therefore, a schematic that shows the role of chemerin with regards to insulin resistance, lipid dysregulation, atherosclerosis, and proinflammatory states would be a useful contribution. Otherwise, this report provides not additional clarity. Could there be any therapeutic insights toward blocking chemerin to its receptor
Thank you for this remark. A mechanism plot for chemerin was added to the manuscript to make it more readable and clear (Figure 1)
Regarding to chemerin blockage, it might have potentially favourable effects, including antiinflammatory effect and and decrease of insulin resistance. There are studies concerning silencing of the CMKLR1 gene. This action was protective against diabetic cardiomyopathy in rats (doi: 10.3389/fphys.2020.00381. eCollection 2020.) or decreased lipid accumulation in mice (doi: 10.1152/ajpendo.00176.2019. Epub 2020 Jan 7.) . However, the correlations that we reported between chemerin and inflammation or insulin resistance parameters were rather weak or moderate. Moreover, chemerin would be rather a biomarker than a causative agent for adverse lipid profile.
Reviewer 2 Report
In this study A. Szpakowicz et al. aim to clarify the association of plasma concentration of chemerin, a relevant an adipokine regulating lipid and carbohydrate metabolism with proinflammatory action, and the presence of CAD with/without diabetes/prediabetes. The main finding, in a small group of patients with stable CAD, is that serum chemerin levels are associated with inflammation markers (WBC count and NLR), insulin resistance and unfavourable lipid profile, but no association is demonstrated with the presence of overt CAD without diabetes/prediabetes, suggesting that although chemerin might be involved in pathogenesis of atherosclerosis on early stages of the diseases it cannot be considered as a biomarker of clinically significant coronary atherosclerosis.
We believe this study provides further evidence that chemerin, alike several molecules of inflammation/lipid metabolism pathways, cannot be considered as a distinctive diagnostic biomarker of coronary artery disease, but rather as a generic marker of inflammation in insulin resistance and obesity as well as in type 2 diabetes mellitus and that its clinical utility is therefore limited.
Besides the serious limitation of the small number of subjects enrolled, especially in Group 2(CAD & normal glucose) n=34 and Group 3(no CAD & normal glucose)n=35, unbalanced as compared with Group 1 (CAD with diabetes/prediabetes, n=129) there are some additional concerns to be appropriately addressed in the discussion/limitations section
1) The authors should clearly state that they are considering only obstructive CAD patients after revascularisation, which represents a subgroup of the most severe stage of stable CAD, requiring PCI : their results cannot be extended to chronic coronary syndrome patients with stable CAD , and even less to subclinical coronary atherosclerosis. This limitation should be underlined.
2) The authors should acknowledge that the unavailability of any coronary imaging and especially CT angiography to rule out CAD in group 3 and adequately characterise atherosclerotic burden in group 1 and 2 is a serious limitation of their study and can profoundly affect their results.
3) The authors state that Group 1 and 2 includes patients with chronic coronary syndrome (CCS) : however, the incidence of angina and dyspnea according to 2019 ESC guidelines definition of CCS is not reported in these patients who underwent PCI in the previous months . Was revascularization uneffective on symptomatology? the authors should discuss this point and define these patients as revascularised stable CAD rather then CCS.
4) The evidence that chemerin is not a prognostic marker in revascularised CAD patients is based on one year follow-up composite event rate of 19/163 pts, probably including stent related complications mainly. We suggest to eliminate this paragraph from the result section because it does not add any clinically relevant information and may be rather misleading .
5) The balance between adiponectin and chemerin serum levels has not been evaluated in this study despite its potential impact on atherogenesis and overt CAD. This should be addressed in the discussion
Author Response
there are some additional concerns to be appropriately addressed in the discussion/limitations section
- The authors should clearly state that they are considering only obstructive CAD patients after revascularisation, which represents a subgroup of the most severe stage of stable CAD, requiring PCI : their results cannot be extended to chronic coronary syndrome patients with stable CAD , and even less to subclinical coronary atherosclerosis. This limitation should be underlined.
Thank you for the comment. We added a paragraph to limitations: Our study group comprised patients with obstructive CAD who underwent PCI in the past, so our results and conclusions cannot be simply extended to patients with CAD and insignificant atherosclerotic lesions that are not qualified for revascularization. Moreover, it is also not representative for subclinical atherosclerosis.
2) The authors should acknowledge that the unavailability of any coronary imaging and especially CT angiography to rule out CAD in group 3 and adequately characterise atherosclerotic burden in group 1 and 2 is a serious limitation of their study and can profoundly affect their results.
This is a very important remark. This issue was already shortly discussed in limitations section: We are not able to decisively comment on the involvement of chemerin in the development of atherosclerosis, since we included patients with established coronary artery disease and the severity of CAD was assessed approximately a year prior to other assessments.
According to your advice we further extended it: Neither invasive coronary angiography was repeated, nor coronary computed tomography angiography was performed at the time of study assessments because of ethical concerns. There were no clinical indications for it and the procedures are associated with risk. Neither type of coronary angiography was performed in a control group, therefore CAD could not be excluded certainly.
3) The authors state that Group 1 and 2 includes patients with chronic coronary syndrome (CCS) : however, the incidence of angina and dyspnea according to 2019 ESC guidelines definition of CCS is not reported in these patients who underwent PCI in the previous months . Was revascularization uneffective on symptomatology? the authors should discuss this point and define these patients as revascularised stable CAD rather then CCS.
According to ESC guidelines 2019 there are 2 forms of CAD: acute and chronic coronary syndromes. Stable patients after revascularization change categorization from acute to chronic. All our patients fulfilled criteria for CCS, which was not clearly stated in the paper and was corrected (lines 78-80).
4) The evidence that chemerin is not a prognostic marker in revascularised CAD patients is based on one year follow-up composite event rate of 19/163 pts, probably including stent related complications mainly. We suggest to eliminate this paragraph from the result section because it does not add any clinically relevant information and may be rather misleading .
Thank you for this remark. Paragraphs concerning prognosis were eliminated from aims, methods, results and discussion.
5) The balance between adiponectin and chemerin serum levels has not been evaluated in this study despite its potential impact on atherogenesis and overt CAD. This should be addressed in the discussion
We described the role of chemerin/adiponectin ratio in the last paragraph of discussion: Some of surprising results of our study could potentially be explained by a novel interesting parameter: chemerin/adiponectin ratio. Unfortunately, adiponectin concentrations were not measured in our study. In previous reports, high values of this ratio were associated with metabolic syndrome or with dyslipidemia (43). (Lines 314-317).
Reviewer 3 Report
This is a clinical study discussing about association between serum chemerin levels and proinflammatory status of adults’ patients. I think the topic is interesting and contributive to risk factor of adults’ proinflammatory status with a clinical study carried wet lab data.
Major concerns:
- Please add a review article or meta-analysis about the similar topic to this study. For example, PMC4859430.
- Based on my knowledge, variance inflation factor should be conducted for collinearity diagnostics of linear regression model. The author should clarify this concern.
- Why the sample size had a difference between three groups? How the author to determine the sample size of each group? The authors should provide a detail about these concerns.
- Please add a mechanism plot about chemerin on proinflammatory status to add readability?
- Please provide detail of stepwise regression model in statistical analysis section.
- In the correlation analysis, the weak association explored in in this study. Could the authors explain this phenomenon?
- The authors may release the data publicly with de-identification for validation, such as Plos one. Would you plan to open a lab data analysis to add readability?
Author Response
Major concerns:
- Please add a review article or meta-analysis about the similar topic to this study. For example, PMC4859430.
Thank you for this suggestion. We added 2 reviews to our manuscript (Reference 5 and 6).
- Based on my knowledge, variance inflation factor should be conducted for collinearity diagnostics of linear regression model. The author should clarify this concern.
Unfotunately VIF is not calculated by Statistica software that was used for analysis. Istead of VIF we are ready to provide R2, which is also a measure of variance. However, providing it togenther with additional p values for all 13 variables that had significant correlation would made the table 4 much less readable. We alredy provided R2 value for multivariable analysis (lines 180-185).
- Why the sample size had a difference between three groups? How the author to determine the sample size of each group? The authors should provide a detail about these concerns.
The study was not designed to investigate associations between 2 specific parameters. The primary analysis included several dozens of variables, so a separate sample size calculation would be needed for each of them. We performed a pilot explorative analysis designed to find associations for further future studies. We understand limitations of this approach, but we also find it reasonable in terms of work management.
- Please add a mechanism plot about chemerin on proinflammatory status to add readability?
A mechanism plot was added (Figure 1)
- Please provide detail of stepwise regression model in statistical analysis section.
A paragraph was added: Multivariable analysis concerning parameters independently associated with chemerin levels was performed with multiple regression. The primary model included WBC count, platelet count, android fat mass, gynoid fat mass, lean body mass, diagnosis of diabetes or prediabetes and c-peptide concentration. Variables were eliminated from the model in a backward stepwise manner. (lines 125-130).
- In the correlation analysis, the weak association explored in in this study. Could the authors explain this phenomenon?
A paragraph was added to Discussion: Many of correlations of chemerin observed in this study are considered weak (R<0.4). We did not expect them to be strong, because atherosclerosis, insulin resistance or inflammmation are very complex processes, influenced by multiple factors. Weak correlations were especially observed in the case of summarized analysis for Groups 1 and 2. When the groups were analyzed separately, the R correlation coefficients were markedly higher. (lines 308-313)
By analyzing separately homogenous groups we limited excessive variance.
- The authors may release the data publicly with de-identification for validation, such as Plos one. Would you plan to open a lab data analysis to add readability?
We fully understand the scope and benefits of data sharing at public depositories. However, in the case of our database, we will be happy to share the individual level patient data on reasonable scientific request.
Additionally, minor spell check was performed.
Round 2
Reviewer 2 Report
The improved version of the manuscript and the authors' replies are good for this reviewer
Reviewer 3 Report
No further comments. Thanks for your efforts on revision.